# Status of Ecosystem Services in Abandoned Mining Areas in the Iberian Peninsula: Management Proposal

**DOI:** 10.3390/toxics11030275

**Published:** 2023-03-17

**Authors:** María González-Morales, Mª Ángeles Rodríguez-González, Luis Fernández-Pozo

**Affiliations:** Environmental Resources Analysis (ARAM) Research Group, University of Extremadura, 06006 Badajoz, Spain

**Keywords:** sphalerite, mining sludge, metal(loid)s, accumulation, *Retama sphaerocarpa*, phytoremediation

## Abstract

An abandoned sphalerite mining area in the southwest (SW) of the Iberian Peninsula was studied to evaluate the impact that the presence of metal(loid)s has on soil and ecosystem health. Five zones were delimited: sludge, dump, scrubland, riparian zone, and dehesa. Critical total levels of lead (Pb), zinc (Zn), thallium (Tl), and chromium (Cr), well above the limit indicative of toxicity problems, were found in the areas close to the sources of contamination. Pb-Zn concentrations were very high in the riparian zone, reaching values of 5875 mg/kg Pb and 4570 mg/kg Zn. The whole area is classifiable as extremely contaminated with Tl, with concentrations above 370 mg/kg in the scrubland. Cr accumulation mainly occurred in areas away from the dump, with levels up to 240 mg/kg in the dehesa. In the study area, several plants were found growing luxuriantly despite the contamination. The measured metal(loid)s content is the cause of a significant decrease in ecosystem services, resulting in unsafe soils for food and water production, so the implementation of a decontamination program is advisable. The plant species *Retama sphaerocarpa*, present in the sludge, scrubland, riparian zone, and dehesa, is postulated as suitable for use in phytoremediation.

## 1. Introduction

Mining has been the economic and social engine of much of the Iberian Peninsula since historical times. However, the decline and abandonment of metal mining from the mid-19th century to the mid-20th century has led to a significant amount of waste being accumulated in dumps and sludge [1]. The inadequate or non-management of these waste materials makes these areas and their surroundings potentially areas of point and/or diffuse pollution [2,3] due to wastewater discharge, acid seepage and emission into the atmosphere of highly particulate materials [4,5], resulting in high anthropogenic pollution [6].

Most metal(loid)s are considered problematic in terms of environmental contamination and toxicity [7,8,9]. Some are biologically essential (manganese (Mn), zinc (Zn), chromium (Cr) or copper (Cu)), and both a deficiency and excess of these can lead to physiological alterations [10,11]. Others such as lead (Pb), cadmium (Cd), mercury (Hg), antimony (Sb) have no known biological function and can be highly toxic to ecosystems and human health [12,13] even at low concentrations [14,15] (Appendix A [16,17,18,19,20,21,22,23,24,25,26]), with the main routes of exposure being inhalation, ingestion and dermal absorption [27].

Lead-Zinc mining is a source of highly toxic metal(loid)s pollution [28], including pollution from arsenic (As), Sb, thallium (Tl), Cd, Hg, Cu, Cr, nickel (Ni), silver (Ag) [29,30,31,32,33]. Lead (Pb) is one of the most abundant and ubiquitous toxic metals associated with mining operations [34]; it is highly persistent in soils [35] and detrimental to human and ecosystem health [36]. In humans, it affects the nervous and circulatory systems and skeletal development and causes endocrine and immune disorders and impaired intellectual development in children [37]. The presence of Zn in soil can alter the enzymatic activities of soil microorganisms, limit the decomposition of organic matter and reduce soil fertility [38,39], but it can be a useful microelement if present in wild plants such as some species of the genus *Aegilops* L., progenitors of cultivated wheat [40]. Elevated levels in the body are associated with tumors, respiratory problems and physiological or pathological alterations [41,42]. The release of Tl into the environment occurs both naturally and anthropogenically, with higher concentrations in the vicinity of metal smelters and coal combustion facilities or in the alteration of sphalerite mineral deposits [43]. Since it is a toxic metal for organisms [44], it is important to monitor its levels in the vicinity of abandoned sulfide ore deposits, where tailings may persist for years.

Assessing the degree of contamination of an environment and the mobility patterns of toxic elements present is vital to establish the risks of dispersion of contaminants through the ecosystem and, when these risks are known, to allow sludge slope stabilization or soil remediation and encapsulation to be undertaken [45,46]. This study refers to various techniques are referred: (i) the formation of a hardpan to facilitate both geochemical reactions and the precipitation of secondary minerals to increase the physical and chemical stability of the sludge [47,48]; (ii) revegetation to reduce water and wind erosion and physically stabilize the sludge [49]; (iii) the application of carbonation and cementation techniques to achieve the physical stabilization and immobilization of released metals [50,51,52] and (iv) the addition of organics, including biochar, compost, biosolids, and organic-rich sludge, to achieve phytostabilization and remediation through the formation of technosoils [53,54,55,56,57,58].

Ecosystem Services refer to the resources provided by the ecosystem; they are the multitude of benefits that nature provides to society. It is therefore very complicated to quantify ecosystem services. In fact, there are many studies that try to establish an economic value for these benefits or for their absence or deterioration [59] through the involvement of specific professional figures in different fields [60]. In this sense, concepts such as soil security try to focus on the fact that the deterioration of the soil resource, because of pollution for example, exerts a negative influence on the ecosystem services of provision, regulation, support or culture [61].

Soil is the main provider of ecosystem services for human survival and environmental maintenance [62]. Among the ecosystem services it provides are agricultural and forest production, protection against erosion and/or flooding, water storage, and atmospheric carbon and nitrogen fixation [63]. These services include hydrological, geochemical and geomorphological processes which are essential for the sustainability of the system [64]. In addition, soil supports all terrestrial life and offers functions that provide environmental goods and services that regulate natural habitat and determine the availability of resources for life, such as food production and water quality [65]. Soil functioning, in terms of its ability to maintain or improve plant productivity and health, as well as its own productivity and health and water quality [62], depends on proper management, as well-preserved soil [66] contributes to a secure food supply [67]. Food security is based on three pillars: availability, access and use, and the final pillar requires, among other factors, the absence of contamination [68].

The Azuaga–Berlanga mining complex, located in the southwest (SW) of the Iberian Peninsula, occupies an area of approximately 100 km^2^, and is the site of an important mining industry that was in operation from the second half of the 19th century until the first half of the 20th century, at which time the facilities were abandoned without any measures to protect human health or ecosystems being adopted. There is a high Pb content in soils near the main town of the mining complex, Azuaga, although there is a lack of other studies related to this problem [69,70].

In this work, the impact of the abandonment of a specific mining operation on soil ecosystem services was evaluated. For this purpose, the concentrations of metal(loid)s (Pb, Zn, Sb, As, Cr and Tl) present in the environment of a former sphalerite (ZnS) mine were studied. The main current land uses were determined and the physicochemical characteristics of the soils, the composition of mining wastes and their geographic distribution over the territory were analyzed to model the origin and distribution of contamination and its impact on vegetation and aquifers. The final objective is to provide tools for the management of abandoned and highly contaminated mining areas.

## 2. Materials and Methods

### 2.1. The Study Area

The study area is in the mining complex called Azuaga–Berlanga in Extremadura in the SW Iberian Peninsula, which includes more than 200 sites of mining operations. The company that carried out the mining was “Sociedad Minera y Metalúrgica de Peñarroya”, which closed all its facilities in the middle of the 20th century. The particular area studied is the area surrounding the San Rafael mine as this is one of the most important Pb-Zn deposits (Figure 1).

The deposit is located south of the Azuaga fault and consists of sandy phyllites with quartzite and greywacke intercalations. It shows hypabyssal mineralization associated with the Lower-Middle Cambrian, whose paragenesis consists of sphalerite (ZnS), galena (PbS), pyrite (FeS_2_), and chalcopyrite (CuFeS_2_), as well as trace minerals which are potentially toxic (Cu or Tl) and quartz and calcite gangue.

The San Rafael mine lies within a small 32-hectare hydrographic basin that defines the study area. Two seasonal streams run through it and flow into the Bembézar River. Today, the remains of constructions—pits, shafts, offices—important tailings dumps, and the mining tailings platform are preserved (Appendix A).

The environment of the mine is an agrosilvopastoral system that in its mature stage is a sclerophyllous forest in the mesomediterranea luso-extremadurense silicícola holm oak (*Quercus rotundifolia*) (*Pyro bourgaeanae*—*Querceto rotundifoliae sigmetum*) series, which, as a consequence of the mining activity has given way to oligotrophic terophytic grasslands and broom, while vallicares and juncales appear in the areas surrounding the seasonal streams [71,72,73].

Five areas were delimited for the study: mining sludge, dump, shrubland, riparian zone, and dehesa (Figure 2).

The mining sludge, corresponding to the mine’s sludge disposal site, comprises a deposit of fine and coarse waste which occupies an area of 2 ha, where the slopes are steeper than 30%. The dump area (3 ha) consists of inert materials deriving from the mine’s solid gangue dump, with a slope of around 5%. The shrubland (4 ha) is a Mediterranean shrub formation, essentially formed by *Retama sphaerocarpa* (L.) Boiss. The riparian zone (3.5 ha) is the vegetation present along the seasonal streams that run through the study area. Finally, the most extensive area (19.5 ha) is the dehesa, which comprises low-tree-density, managed oak woodland with little associated scrub which is put to agrosilvopastoral use based on extensive livestock farming and hunting activities (Figure 2). The dominant soils are Eutric Regosols [74], with a low organic matter content and a light texture. The relief presents a slightly eastwards-sloping topography.

The region has a Mediterranean climate with Atlantic influence, low rainfall (average 514 mm/yr.), and marked thermal variations between winter and summer (8 °C and 26 °C). Winters are short, but with intense frosts. Spring and autumn temperatures are mild. Annual evapotranspiration is 884 mm, so that the area’s aridity index corresponds to dry subhumid [75].

### 2.2. Sampling

Dump (3), soil (37), mining sludge (3), and vegetation (163) samples were taken. Soil and mining sludge samples were taken in the first 30 cm with an auger, and dried at room temperature, sieved at 2 mm and ground and sieved at 0.25 mm. Around these samples, in a radius of 5 m, the aerial part of the predominant vegetation in that area was collected and three vegetation habit categories were differentiated—tree, shrub, and grass. In the laboratory, the vegetation was washed with distilled water and dried in an oven at 60 °C, then ground and sieved to 0.25 mm. Soil and vegetation samples were also collected in an area far from the mining site, located 5 km to the E of the study area, with identical pedo-environmental characteristics, but free of contamination. These samples were taken as a control.

### 2.3. Mineralogical Characterization of Dump and Soils

The mineralogical composition of the dump and soils was determined by X-ray diffraction (XRD) using the disordered crystalline powder method. The diffractograms were recorded on a Bruker D8 ADVANCE diffractometer. Copper Kα_1_ radiation (λ = 0.15406 nm) was used, scanning the 2θ angular range from 5° to 70°.

### 2.4. Physicochemical Characterization of Dump, Soils, Plant Tissues, and Mining Sludge

Only the bulk density was determined in soils in the field [76]. The rest of the analytical parameters were determined in soils and mining sludge. pH [77] and electrical conductivity [78] were found using a Hach sensION™ + pH3 Lab Meter and a Hanna model HI99301 conductivity meter, respectively. The carbonate content was determined using a Bernard calcimeter [79]. Organic carbon was determined by dry combustion −950 °C- with oxygen in a TruSpec Micro (LECO) macro-sample elemental analysis instrument. Cation exchange capacity (CEC) was determined by saturation with ammonium, removal of the excess, andlater substitution with ammonium [80]. The texture was determined using the Bouyoucos method [81]. To determine the total metal(loid)s in soil, plant tissues, and the water soluble fraction, the samples were sieved at 0.25 mm and analyzed by ICP-MS (inductively coupled plasma mass spectrometry) after digestion with HCl and HNO_3_ in a microwave oven [82], using an Agilent Tech model 7900 system. Since this method is not intended to accomplish total decomposition of the sample, the extracted analyte concentrations of the dump, soils, and mining sludge may not reflect the total content in the sample. Hydraulic conductivity in dump and mining sludge was determined using a laboratory permeameter and applying Darcy’s law [83], as was the content of metal(loid)s in the leachate.

### 2.5. Data Analysis

#### 2.5.1. Soil Accumulation Index

The geoaccumulation index (I_geo_) [84], used to quantify the level of a soil’s heavy metal contamination is
(1)$$I_{\mathrm{geo}}={\mathrm{Log}}_{2}\left(\frac{C_{n}}{1.5B_{n}}\right),$$
where:

C_n_ is the concentration (mg/kg) of metal in the soil, and B_n_ is the geochemical background for the metal [85]. The values are classified as follows: I_geo_ ≤ 0 uncontaminated; 0 < I_geo_ ≤ 1 uncontaminated to moderately contaminated; 1 < I_geo_ ≤ 2 moderately contaminated; 2 < I_geo_ ≤ 3 moderately to heavily contaminated; 3 < I_geo_ ≤ 4 heavily contaminated; 4 < I_geo_ ≤ 5 heavily to extremely contaminated; I_geo_ > 5 extremely contaminated.

#### 2.5.2. Generic Reference Levels

In the Extremadura Region, the threshold concentrations from which the contaminants cause damage to human or ecosystem health are set in accordance with the values given in the current legislation on soils [86]. These are termed the “Generic Reference Levels” (henceforth NGR, to use the legislation’s Spanish acronym). However, in the case of some heavy metals (Tl) the NGR value is not covered by regional legislation, and for these we have used the Canadian Environmental Quality Guideline [87].

#### 2.5.3. Spatial Distribution of the Contamination

From the I_geo_ data, the spatial distribution of the contamination was modeled using ArcGIS software. A semivariogram of the data was constructed, followed by ordinary kriging [88,89]. Maps were constructed using ArcMap v. 10.7.1 [90].

#### 2.5.4. Statistics Analysis

The descriptive statistical study and the correlation factor analysis for the elaboration of a principal component matrix, with Varimax rotation and Kaiser normalization, was carried out using SPSS Statistics v.23 software. Data tabulation and basic calculations of means, standard deviations and proportions were performed using Excel software [79].

#### 2.5.5. Bioaccumulation Factor (BF)

Using the bioaccumulation factor (BF), which is the ratio of metal(loid) concentration in the plant (mg/kg) to the water soluble metal(loid) concentration in the soil (mg/kg) [91], the soluble Pb, Zn, Sb, As, Cr, and Tl content in soils was used to estimate the ability of the plant species to accumulate these elements.

#### 2.5.6. Vegetal Cover

The data on the determination of the vegetation cover were obtained from LIDAR information from the CNIG (National Center for Geographic Information), 2nd coverage 2015-present [92]. QGIS 3.22.12 software was used [93].

## 3. Results

Mineralogical analysis of the dump mainly revealed the presence of quartz (SiO_2_); alkali feldspars (orthoclase, sanidine, and albite); and small amounts of calcite (CaCO_3_). Valentinite (Sb_2_O_3_), galena (PbS), and iron-rich sphalerite (ZnFe)S were also detected, as well as secondary products of these minerals, such as cerussite (PbCO_3_), litharge (PbO), and wurtzite (ZnS). The mineralogy of the soils is mainly quartz, feldspars (orthoclase, sanidine, and albite) and various clay minerals (illite, muscovite, and chlorite).

The predominant soil type is Eutric Regosol. In none of the areas studied did the bulk density (BD) indicate any risk of compaction (Table 1). The soils and sludges have a pH close to neutral, especially those near the dump. Organic carbon contents were low to medium (6.13–21.70 g/kg), corresponding to what is usually found in nearby uncontaminated soils [94]. The soils have a loamy texture with the sand fraction being predominant, a low Cation Exchange Capacity (CEC), and electrical conductivity (Ce) since none of the study areas present significant salt content (Appendix A). The carbonates content is minimal except in the scrubland zone, where it exceeds 4%. Finally, the hydraulic conductivity (k) in the sludge and dump is 9.9 × 10^−7^ m/s and 9.9 × 10^−6^ m/s, respectively.

Table 2 shows the metal(loid)s that presented significant concentrations, showing a large variability.

Figure 3 shows a comparison of the metal(loid) concentrations with the applicable NGR in each case. For the assignment of the corresponding NGRs, the sludge is considered to be an industrial-use zone, the scrubland and riparian zone are considered to be ecosystems, and the dehesa and control zones, since they are dedicated to tillage, livestock, and/or hunting activities, are considered to be other use zones. The highest concentrations of Pb and Zn are found in the sludge, with much lower values being found in the scrubland and dehesa. Although Sb follows the same pattern, its concentrations are much lower. However, the behavior of As and Cr is the opposite, with the highest concentrations being found at the points farthest from the sludge, with Cr concentrations being higher than those of As. The Tl content is very high in all zones. The samples corresponding to the control zone show low concentrations of all the metal(loid)s studied except for Cr and Tl.

Principal Component Analysis (PCA) of the elements established two principal components with values explaining more than 80% of the variation. The first includes Cd, Zn, Pb, iron (Fe), and Sb and the second As and Tl.

### 3.1. Spatial Distribution of the Contamination

There are very wide variations in I_geo_ (Figure 4); the spatial distribution suggests contamination by Tl, Pb, and Zn throughout the study area. Thallium contamination is extreme, and lead and zinc contamination is extreme in the sludge (1), dump (2), and towards the east of the riparian zone (4).

### 3.2. Vegetation

Figure 5 shows the vegetation cover in our study area, highlighting a ground cover of around 80% in the dehesa, mainly due to the presence of trees. In the rest of the formations the vegetation cover is scarce, with values between 5 and 11% in the riparian and shrubland formations. The vegetation present in the delimited zones of our study area presents moderate to high contents of the analyzed metal(loid)s (Figure 6).

Table 3 shows the bioaccumulation factor (BF) in each of these zones. Thallium (Tl) presents a very high BF in all the plant species present; Cr and Pb in *Q. rotundifolia* Lam.; Cr and Zn in *R. sphaerocarpa* (L.) Boiss, and Cr, Pb and Zn in *S. holochoenus* (L.) Soják and *Carlina* ssp.

## 4. Discussion

The soil characteristics of the study area coincide with those reported in previous works in the area [94,95,96]. Hydraulic conductivity (k) coupled with the low CEC (12.7 Cmol/kg), together with the characteristics of the sludge and soils, indicate a higher concentration of metal(loid)s in the leachates generated in the sludge; however that high concentration of metal(loid)s in the leachates would not modify particle aggregation or soil structure [97]. The low CEC may be due both to the low colloid content of the soil and to the nature of the clay minerals present (chlorite and illite). A similar situation occurs in soils developed under similar environmental conditions [94]. It should also be noted that the edaphic parameters analyzed did not differ between the study area and the control area, only the metal(loid) content differed (Table 1 and Table 2).

The weakly acidic character and low carbonates, CEC, and colloid content of the soils favor the mobility of contaminant elements [97]. It has been shown [98,99] that the accumulation of metal(loid)s in a soil is controlled by processes such as adsorption, sorption, and/or complexation in colloids (clay minerals, iron, and/or manganese oxides, and organic matter) as well as by co-precipitation with other elements.

Eutric Regosols are poorly developed soils, either due to little alteration of the original material or to intense degradation because of erosive processes and, in these cases, contamination processes can cause more damage to the ecosystem services of the territory because of diffuse contamination, a conclusion that agrees with other studies [100].

The correlation between elements provides important information on their origin and dispersion [101]. In our case, Pb, Sb, Zn, and Cd show a correlation index higher than 90%. This high significance, together with the results obtained in the PCA, indicates that these elements have the same origin and/or dispersion pathway (Appendix A and Appendix A).

The wide range of minority elements and trace elements that always accompany Pb-Zn deposits have been leached and transported by water. The percolation study conducted shows that the leachate from the sludges, consisting of very fine-grained residues with very slow k, contributes large amounts of metal(loid)s (Table 2).

The absence of Pb-Zn in the mineralogical study of the soils is indicative of strong chemical weathering of the sphalerite and galena mining residues. The wide range of minority and trace elements that always accompany these Pb-Zn deposits were leached and transported by water, hence the significant concentrations of Pb, Zn, Fe, Sb, Cr, and Tl, with Zn, Pb, and Tl being the most abundant in the vicinity of the mine.

The highest concentrations of Tl were detected in the scrubland, decreasing towards lower topographic levels (riparian zone and dehesa). Some authors [102,103] have pointed out that the presence of Tl in a soil is related to its clay content. Maluszynski (2009) [104] suggests that a low organic matter content, an alkaline character, and the presence of illite and chlorite (minerals with a moderate alteration capacity) do not favor strong Tl retention, coinciding with the characteristics of the soils studied.

The halo of Pb and Zn contamination extends in a southwest—northeast direction, around the sludge and dump areas, following the runoff slope of the terrain. The cohesive nature of the sludge materials, together with slopes exceeding 30%, favor erosion and washing towards the seasonal stream. The behavior of Sb is the same, although its concentrations are lower.

The main anthropogenic sources of Pb, Zn, and Tl dispersion are the extraction of metal sulfides and their treatment by flotation, sludge and smelting [104]. In the present case, the main source of Pb-Zn contamination was detected in the dump and sludge areas, although it greatly exceeded the corresponding NGRs in almost all areas. Concentrations in the scrubland, and especially in the riparian zone (120 times higher than the NGR for Pb and 16 times for Zn), are of concern, particularly because these areas follow the course of seasonal streams that flow into the Bembézar river. This river supplies irrigation water to the surrounding agricultural areas and for many years has been used to supply drinking water to nearby towns. The river is dammed and is considered an ecological corridor [105,106]. The high concentration in the riparian zone is attributable to water erosion of the materials by runoff from the dump and sludge. In the dehesa, however, Zn concentration is no longer problematic.

The Pb-Zn levels found at the San Rafael mine are higher than those referred to in the literature for other abandoned mining areas within the Azuaga–Berlanga formation: 21–3275 mg/kg [69] or 42–179 mg/kg Pb and 63–156 mg/kg Zn [70]. Values reported for a pyrite deposit located in southern Portugal were 38–3500 mg/kg Pb and 39–945 mg/kg Zn [107]; in a Pb-Zn mining area in northern Morocco in 1935 values were mg/kg Pb and 47 mg/kg Zn [108]. Values higher than those obtained in the present study have only been described for a Pb-Zn mine dump in southern Poland, where concentrations of 32,193–11,9147 mg/kg Pb and 17,331–67,763 mg/kg Zn were found [108]. Although Pb-Zn concentrations in the works cited are lower than those described by us, this may be because there are no sludge deposits in the study area of those works, with the exception of the area in southern Poland [109].

Thallium levels were high regardless of the area studied and proximity to the source of contamination. According to the I_geo_ values, the whole study area is extremely polluted, greatly exceeding the NGR in all zones, with the lowest levels relating to the dehesa. These values are higher than those reported in the mining waste areas, where Tl concentrations of 35–513 mg/kg are reached [109]. The fact that Tl is distributed throughout our study area makes us suspect its presence is due to a continuous process of diffuse contamination, either by alteration of sphalerite [110] or as an emission in coal combustion in power plants [111]. The dumps and sludges which are present, as well as the operation of a coal-fired power plant from 1976 to 2020 within 50 km [112] of the study area, may be responsible for the Tl presence in our study area. A similar situation is reported by other authors [113]. Despite its toxicity, Tl is not very mobile, and although in this work the speciation of the elements present has not been studied, it is likely that the chemical species involved is Tl (I) since it has been detected (data not published) by Laser-Induced Breakdown Spectroscopy (LIBS) at a wavelength of 323 nm.

The concentration of Cr exceeds the NGR in all areas, with the dehesa being the most contaminated, and the consequent deterioration and damage to activities being due to the agricultural activity that takes place there since the use of agrochemicals has been shown to be the reason for the presence of Cr in agricultural soils [114].

According to the applicable legislation, the soils in the study area are contaminated by Pb, Zn, Cr, and Tl since the concentrations of these metals exceed the NGR in all cases. However, this statement should be interpreted with some caution since the regulations only consider the total concentrations of these metals, ignoring attributes such as their speciation, bioavailability, and mobility that can lead to overestimating or even underestimating the risk.

The abandonment of mining activity has caused pollution of the environment, as evidenced by the I_geo_. This pollution has been accentuated by the pedo-environmental characteristics of the territory and by the lack of any protective measures, with runoff being the main cause of the spread of pollution.

The source of the contamination is the sludge and the dump. As can be seen in Figure 2, vegetation cover is almost non-existent (3.6% and 0.7% in the dump and sludge, respectively) and this low cover means that there is a maximum contribution of toxic elements to the rest of the areas. This situation leads to the degradation of soil health and, therefore, has a negative impact on its ecosystem services. These services are unsafe for food and water production. Nevertheless, plant communities have flourished throughout the area (Table 4), suggesting the need for their further study.

The high concentrations of Pb, Zn, and Tl found imply a significant risk to the health of the ecosystem since vegetation does not absorb and accumulate these metals randomly but selectively, as natural vegetation responds to the types and quantities of contaminants [115] thus increasing the risk of them being incorporated in the trophic chain. In this way, the metals could concentrate in organisms. Vegetation cover is high in the dehesa, close to 80%, although this is mainly due to the presence of trees. In the riparian zone and, above all, in the scrubland, the vegetation cover is scarce.

The results have allowed us to quantify and model the processes derived from the current situation and this can serve as a guide for the development of reclamation programs for abandoned mining areas.

The high concentration of Tl in all areas, and in turn in all vegetation, will require further study, as it is surprising that all plants show high or extremely high BF. As indicated, possible diffuse contamination by deposition from coal burning could influence the presence of Tl in the vegetation.

As for the rest of the elements and the vegetation in the different established areas, *Q. rotundifolia* Lam., *R. sphaerocarpa* (L.) Boiss, *S. holochoenus* (L.) Soják, and *Carlina* ssp. L. not only resist the present contamination, but also present phytoextraction capacity, evidenced by the BF. *Carlina* ssp. is a nitrophilous annual species that often occurs in livestock areas due to the feces [116]. *S. holochoenus* (L.) Soják requires humid environments to grow, hence it being primarily present in riparian formations, and less frequently present in scrublands. *R. sphaerocarpa* (L.) Boiss is a legume which grows in a Mediterranean climate and which has a deep root system [117] and N-fixing capacity due to symbiosis with rhizobia nodules in its root system. It is able to colonize surfaces subjected to high environmental stress [118]. These circumstances motivate us to propose *R. sphaerocarpa* (L.) Boiss phytoextraction or phytostabilization of the studied metal(loid)s, especially Cr and Zn.

The strategy proposed for soil decontamination and the consequent recovery of ecosystem services is the stabilization of the sludge and dump, since these are the origin of contamination and associated toxicity, especially that due to Pb, Zn, Cr, and Tl. Subsequently, revegetation with native vegetation, mainly *S. holochoenus* (L.) Soják in the riparian formation and *R. sphaerocarpa* (L.) Boiss in the other formations, is suggested as a measure for landscape restoration, improvement of the physicochemical properties and decontamination of soils.

## 5. Conclusions

This study shows the negative impacts of metal(loid)s on ecosystem services and soil security as a result of the dumps and sludge generated by mining activity being abandoned. This is evidenced by indices such as I_geo_ and NGRs, with the water and food supply being the two ecosystem services most heavily affected.

Therefore, it is necessary to implement management programs in mining areas to prevent contamination from entering the food web and to guarantee safe soils for food and water production. These programs should be developed based on an adequate management of the characteristics of both the territory and the source of the contamination and should include (i) soil stabilization for effective erosion control and a reduction in the risk of causing diffuse contamination processes, (ii) protection of water resources, and (iii) establishment of a cover of native vegetation that can act as a phytoremediation agent in the area.

The main source of soil contamination is the high presence of metal(loid)s in the runoff originating from the sludge and dump. The accumulation of metal(loid)s in nearby areas where agricultural, forestry, grazing, and hunting activities take place has adverse effects on both the ecosystem and human health. High concentrations of Pb and Zn are found particularly in the riparian zone, in streams that, although seasonal, flow into a larger river, with consequent adverse effects on the quality of water used for irrigation and human supply. Since scrubland is the most abundant plant formation in the study area, *Retama sphaerocarpa* (L.) Boiss would be a good candidate for soil remediation.

## Figures and Tables

**Figure 1 toxics-11-00275-f001:**
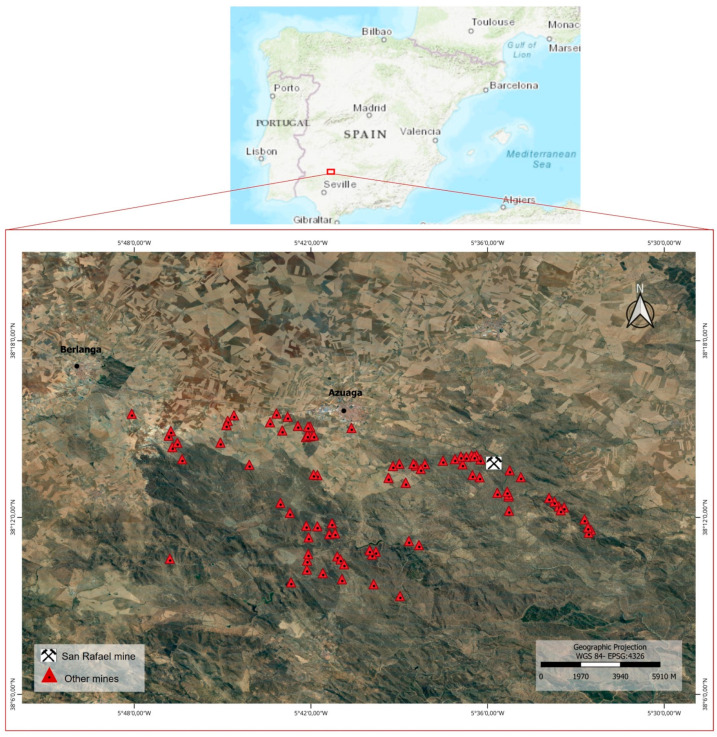
Location of the San Rafael mine (Azuaga, Spain).

**Figure 2 toxics-11-00275-f002:**
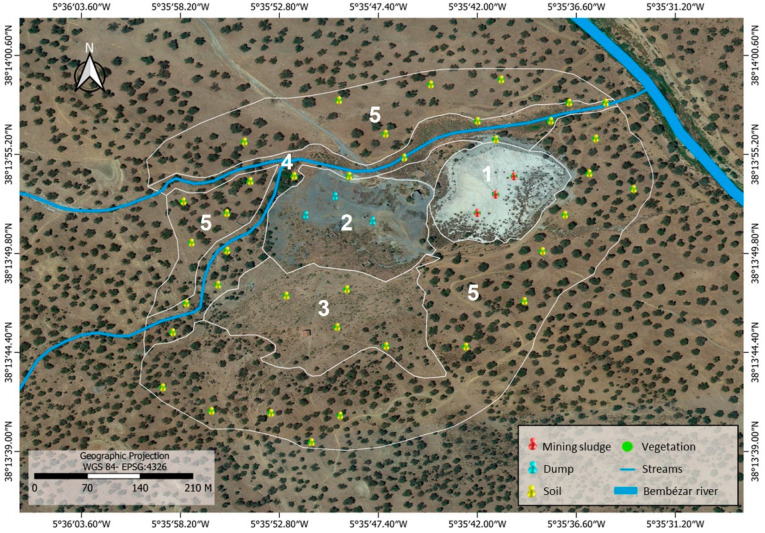
General map of the study area and sampling points: (1) mining sludge, (2) dump, (3) shrubland, (4) riparian zone and (5) dehesa.

**Figure 3 toxics-11-00275-f003:**
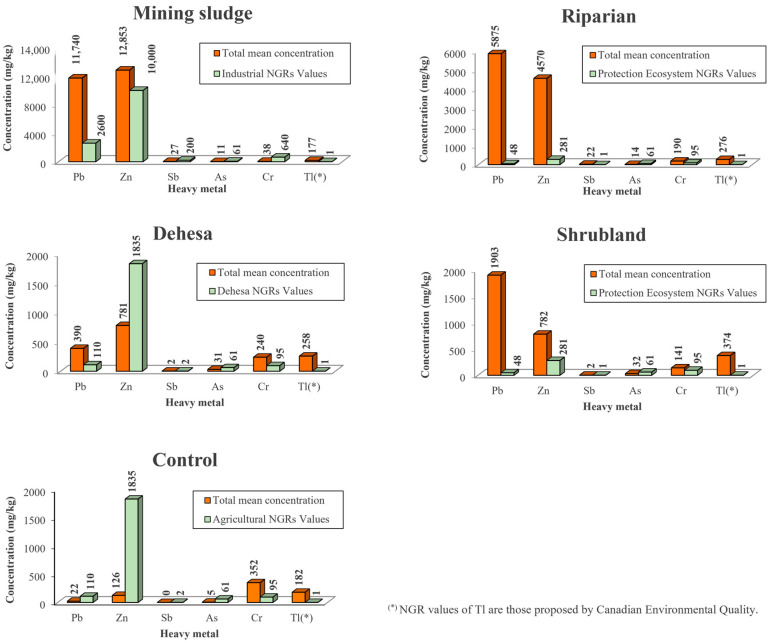
Total mean concentrations of the metals studied versus values for the NGR (Generic Reference Levels) in each of the zones. Mining sludge (n = 3), shrubland (n = 4), riparian (n = 12), dehesa (n = 21), control (n = 3).

**Figure 4 toxics-11-00275-f004:**
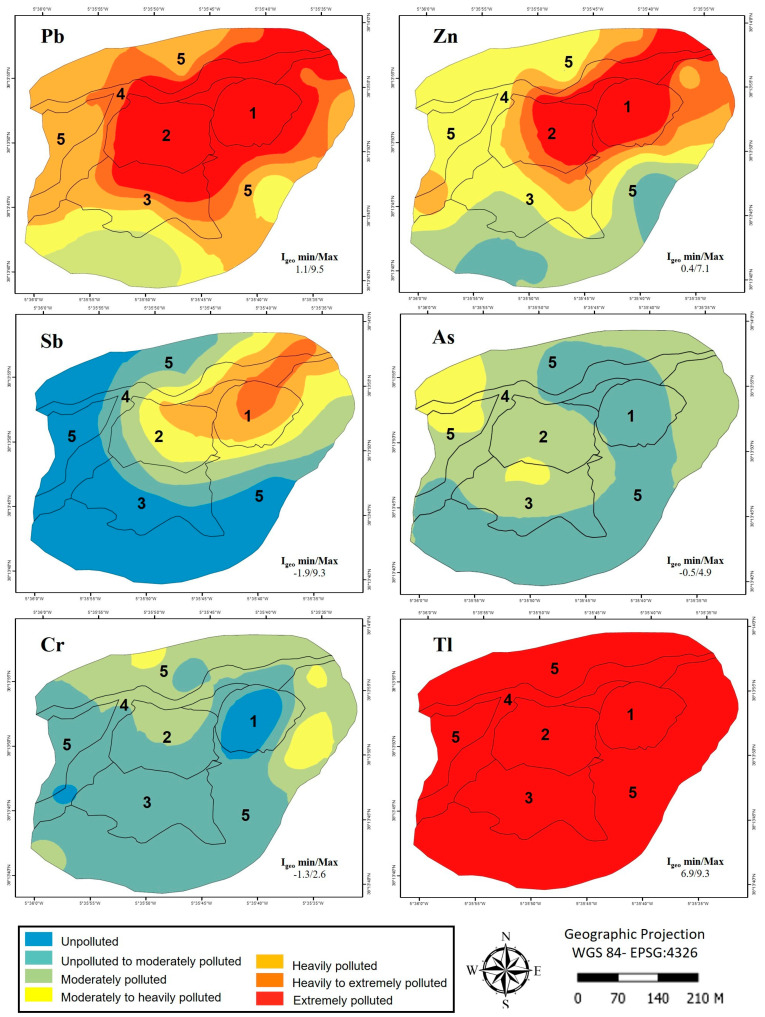
Pollution distribution according to the I_geo_ in the study area: (1) mining sludge, (2) dump, (3) shrubland, (4) riparian zone, and (5) dehesa. Soil (n = 37).

**Figure 5 toxics-11-00275-f005:**
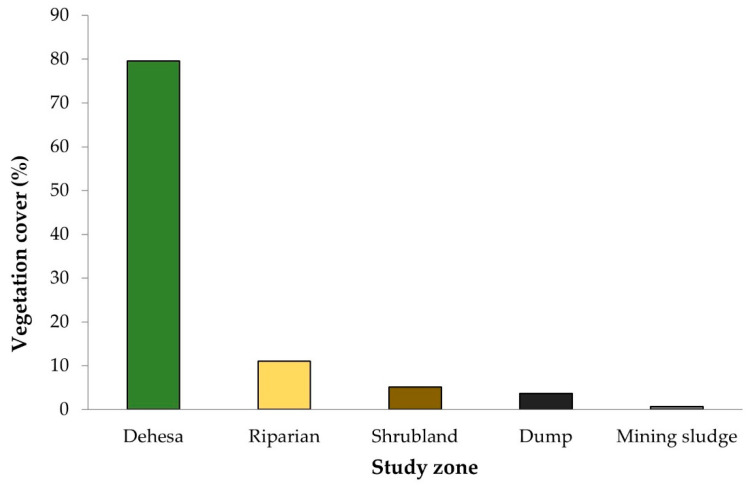
Vegetation cover.

**Figure 6 toxics-11-00275-f006:**
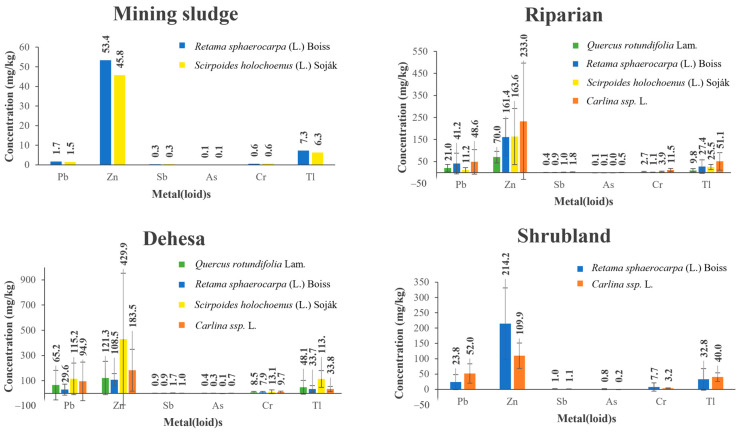
Average concentration of metal(loid)s in vegetation. Mining sludge (n = 4), riparian (n = 29), dehesa (n = 49), shrubland (n = 8).

**Table 1 toxics-11-00275-t001:** Physical–-chemical characteristics of the soils and sludge collected at the San Rafael mine. Mining sludge (n = 3), shrubland (n = 4), riparian (n = 12), dehesa (n = 21), control (n = 3).

Zone	BD (Mg/m^3^)	pH	EC (dS/m)	Carbonate (gr/kg)	Organic C (gr/kg)	CEC (Cmol/kg)	Texture
Mining sludge	-	6.7 ± 0.2	0.60 ± 0.05	12.9 ± 4.8	6.13 ± 0.48	1.06 ± 0.08	Silt loam
Shrubland	1.34 ± 0.09	6.6 ± 1.1	0.10 ± 0.01	38.3 ± 23.3	21.70 ± 5.08	9.29 ± 0.19	Loam
Riparian	0.92 ± 0.12	6.3 ± 0.3	0.20 ± 0.02	10.9 ± 2.8	21.27 ± 6.93	12.83 ± 2.88	Sandy loam
Dehesa	1.37 ± 0.22	5.9 ± 0.5	0.10 ± 0.01	19.6 ± 9.5	15.16 ± 2.70	9.95 ± 2.40	Sandy loam
Control	1.93 ± 0.66	6.51 ± 0.26	0.10 ± 0.01	-	8.53 ± 1.79	12.51 ± 1.74	Sandy loam

BD: Bulk density; EC: Electrical conductivity; CEC: Cation exchange capacity.

**Table 2 toxics-11-00275-t002:** Average concentration of metal(loid)s in mining sludge, soils, and leachates. Mining sludge (n = 3), shrubland (n = 4), riparian (n = 12), dehesa (n = 21), control (n = 3).

Zone	Pb	Zn	Sb	As	Cr	Tl	Fe	Cd
	mg/kg							
Mining sludge	11,740 ± 6578	12,853 ± 5005	27 ± 4	11 ± 1	38 ± 1	177 ± 75	33,214 ± 1748	2 ± 1
Shrubland	1903 ± 2677	782 ± 345	2 ± 2	32 ± 22	141 ± 21	374 ± 76	72,817 ± 6650	n.d.*
Riparian	5875 ± 5522	4570 ± 4008	22 ± 25	14 ± 3	190 ± 91	276 ± 58	68,976 ± 9322	n.d.*
Dehesa	390 ± 342	781 ± 631	2 ± 2	31 ± 50	240 ± 128	258 ± 73	70,498 ± 10,834	n.d.*
Control	22 ± 1	126 ± 8	n.d.*	5 ± 1	352 ± 247	182 ± 15	81,420 ± 4416	n.d.*
Leached	µg/L							
Mining sludge	163.1 ± 26.4	454.2 ± 353.3	2.5 ± 0.3	n.d.*	2.5 ± 0.6	-	537.6 ± 48.5	15.8 ± 1.7
Dump	5.4 ± 1.8	155.3 ± 4.5	n.d.*	n.d.*	n.d.*	-	n.d.*	n.d.*

(*) n.d.: Not detected.

**Table 3 toxics-11-00275-t003:** Bioaccumulation factor. Mining sludge (n = 4), riparian (n = 29), dehesa (n = 49), shrubland (n = 8).

Mining Sludge	*Quercus rotundifolia* Lam.	*Retama sphaerocarpa* (L.) Boiss	*Scirpoides holochoenus* (L.) Soják	*Carlina* ssp. L.
Pb	-	0.00	0.00	-
Zn	-	0.02	0.01	-
Sb	-	0.03	0.03	-
As	-	0.02	0.02	-
Cr	-	2.18	2.18	-
Tl	-	4.20	3.62	-
Shrubland				
Pb	-	0.30	-	0.66
Zn	-	2.32	-	1.19
Sb	-	0.08	-	0.08
As	-	0.04	-	0.01
Cr	-	2.91	-	1.21
Tl	-	504.62	-	615.38
Riparian				
Pb	0.01	0.02	0.01	0.03
Zn	0.01	0.03	0.03	0.04
Sb	0.02	0.04	0.05	0.09
As	0.00	0.00	0.00	0.02
Cr	0.61	0.25	0.89	2.62
Tl	49.41	138.15	128.57	257.65
Dehesa				
Pb	1.84	0.84	3.26	2.69
Zn	0.83	0.74	2.93	1.25
Sb	0.11	0.11	0.21	0.12
As	0.03	0.02	0.01	0.04
Cr	2.53	2.35	3.90	2.89
Tl	890.74	624.07	2103.70	625.93

**Table 4 toxics-11-00275-t004:** Number of plant species found around the sampling points by habit and by zone.

Zone	Tree	Shrub	Grass	Total
Mining sludge	0	1	2	3
Dump	0	0	1	1
Shrubland	1	1	5	7
Riparian	1	4	10	15
Dehesa	1	2	10	13
Control	1	1	1	3
Total	1	5	13	

## Data Availability

Not applicable.

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
