# Peer review of "Status of Ecosystem Services in Abandoned Mining Areas in the Iberian Peninsula: Management Proposal"

_toxics, 2023, doi:10.3390/toxics11030275_

Round 1

Reviewer 1 Report (New Reviewer)

The authors propose a manuscript titled “Status of Ecosystem Services in abandoned mining areas. Management proposal

The article is original, well structured and written. In particular, this study takes into consideration the An abandoned sphalerite mining area in the SW of the Iberian Peninsula was studied in order to verified the presence of metal has on soil and the ecosystem health in 5 different zones. The Pb, Zn, Tl and Cr, are the heavy metal present with high level of concentration and in particular Pb-Zn concentrations were very high in the riparian zone, and the whole area have as extremely contaminated in Tl especially in the scrubland, while the Cr accumulation occurred mainly in areas away from the dump in dehesa habitat. Many plants were found growing luxuriantly despite the contamination. The authors declare correctly that the presence of these metals contents are the cause of a significant decrease in ecosystem services, and is not evaluatable the food and water production, before the decontamination actions. Retama sphaerocarpa has benn choosed for some habitat as the the best plant species for use in phytoremediation.

I read the work carefully in a critical way, and suggested in detail some crucial concepts only in order to further improve the good work done (in bold). In particular, some concepts need to be referenced, indicating point by point where is necessary.

Title. Status of Ecosystem Services in abandoned mining areas of Iberian Peninsula: management proposal

1.      Introduction

Well done, the concept are correct but in some case need to complete in the correct way. The suggestions are in bold:

·         Lines 23-25. …However, the decline and abandonment of metal mining from the mid-19th century to the mid-20th century has led to a significant amount of waste accumulated in dumps and sludges. [choose a reference];

·         Lines 41-43. …The presence of Zn in soil can alter the enzymatic activities of soil microorganisms, limit the decomposition of organic matter and reduce soil fertility [27,28], but it can be a useful microelement if possessed in wild plants such as in some species of the genus Aegilops L., progenitors of cultivated wheat [Perrino et al. 2014]”;

·         Lines 61-64. …Ecosystem Services refer to the resources provided by the ecosystem, they are the multitude of benefits that nature provides to society, therefore the quantification of them is very complicated, in fact many are the studies that try to establish an economic value to these benefits or to their absence or deterioration [47], through the involvement of specific professional figures in different fields [Pisani et al. 2021]”;

References to be added

ü  Perrino, E.V.; Wagensommer, R.P.; Medagli, P. The genus Aegilops L. (Poaceae) in Italy: taxonomy, geographical distribution, ecology, vulnerability and conservation. Systematics and Biodiversity 2014, 12, 331-349. https://doi.org/10.1080/14772000.2014.909543

ü  Pisani, D.; Pazienza, P.; Perrino, E.V.; Caporale, D.; De Lucia, C. The Economic Valuation of Ecosystem Services of Biodiversity Components in Protected Areas: A Review for a Framework of Analysis for the Gargano National Park. Sustainability 2021, 13, 11726. https://doi.org/10.3390/su132111726

2. Materials and methods

·         Figure 1. The map is well done, but in the text is not specify the geographical system used, WGS84? Please specify. Also in the map the authors used the metric coordinates, while in the text is different 5º 35’ 45.07” W and 38º 13’ 53.07” N. Please standardize;

·         Figure 2. The geographic system is different then Figure 1? Please standardize. Also is pleonastic reporting W and N for longitude and latitude;

·         Lines 137-138. The authors declare: “the aerial part of the predominant vegetation in that area was collected and three vegetation habit categories were differentiated – tree, shrub, and grass –“. The phytosociological method is usually applied when we speak of vegetation surveys. Just in Spain Rivas-Martinez, who died a few years ago, worked a lot for the Mediterranean Basin and for Spain.

3. Results

·         The figures and tables are clear, well done.

·         3.2 vegetation. See my previous comment, I think is too poor reporting only these information on vegetation, as Q. rotundifolia R. sphaerocarpa, S. holochoenus. Do you have other specific information at least on syntaxa? See: Loidi J. 2012. The Vegetation of the Iberian Peninsula. Volume 12 https://link.springer.com/book/10.1007/978-3-319-54784-8

4. Discussion

Well done, only one crucial point

·         Lines 367-368. See my previous comment on the crucial point on vegetation type detected. The high concentrations of Pb, Zn and Tl found imply a significant risk to the health of the ecosystem since vegetation can absorb and accumulate these metals, not randomly but selectively as natural vegetation responds in relation to the types and quantities of contaminants (Perrino et al. 2014);

·         Lines 380-381. When report for the first time the scientific name of plant species, please remember to report also the name of the author that classify the species. I suggest this website http://ww2.bgbm.org/IOPI/gpc/query.asp

Ø  Q. rotundifolia Lam.

Ø  R. sphaerocarpa…

Ø  S. holochoenus…

Reference to be added that I suggest to read

ü  Perrino, E.V.; Brunetti, G.; Farrag, K. Plant communities of multi-metal contaminated soils: a case study in National Park of Alta Murgia (Apulia Region - southern Italy). International Journal of Phytoremediation 2014, 16, 871-888. https://doi.org/10.1080/15226514.2013.798626

Please follow the guidelines of the journal about references

Author Response

Dear Referee, we would like to thank the referees for their suggestions, which we have incorporated into the text. We believe that their valuable contributions have enriched the work.

Reviewer 2 Report (New Reviewer)

I read the manuscript interestingly and the authors have worked on very important question. However, there are still some issues that have to be addressed by the authors before considering the manuscript for publication.

1- Authors should review the title. The title is not informative about the goals or results of the study. Judging by the title, the paper appears to be a review about ecosystem services in mines in general. Which is not the case. The title is also misleading, with the appearance that several mines are discussed ("...in abandoned mining areas"), when in fact only one mine is studied. The title should be concise, but should reflect the study that was carried out.

2- Throughout the text: All plant species names should be accompanied by the botanical authority at first mention. In this way, if a plant name is changed in time, it will still be possible to recognize it. Be constant in including botanical authorities when a species is mentioned for the first time.

3- Fig.1: All maps (and satellite images) must indicate scale.

4- The authors refer to “total metal(loid)s soil”. However, they used a digestion method with nitric acid and hydrochloric acid. Therefore, no hydrofluoric acid was used, so the silicates were not dissolved and therefore the measured concentrations are not total. They are only pseudo-totals. This is the EPA Method 3051 (or a similar). In fact, if the authors read carefully the description of the method, is mentioned the following: “Since this method is not intended to accomplish total decomposition of the sample, the extracted analyte concentrations may not reflect the total content in the sample.” Therefore, if the authors intended to obtain the total concentrations, the samples digestion method was poorly chosen.

5- The “2.4. Physicochemical characterization of dump, soils and mining sludges” section should also give more details about Quality Assurance and Quality Control. Authors should indicate the obtained accuracy values. Were used reference materials? If so, these should be listed. It would be also interesting to provide the reader with limits of detection/determination of analyzed elements.

6- The authors refer to “Bioaccumulation factor”. However, they say nothing about the chemical analysis of plants. How were the samples digested? How were they analyzed?

7- Line 192: “Soluble Pb, Zn, Sb, As, Cr and Tl contents in plants leaves were used…” - please explain.

8- Tables: Authors should indicate the number of samples (n =). What is the meaning of the ± (error? Standard deviation?).

9- Figures: Authors should indicate the number of samples (n =).

10- Fig. 3 and Fig. 6: All results of concentration in analyzed samples given in figures, as well as tables and in text of manuscript should be given with measurement uncertainty (or standard deviation).

11- Fig. 6 and Table 3 (and throughout the text): In the spelling of the scientific names of the species, the binomial nomenclature rules should be applied always! Both the first part of the name, the genus, and the second part, the species, should be italicized when a binomial name occurs in normal text, but the botanical authority not.

12- Throughout the manuscript: Some minor typos, grammar and syntax errors should be carefully revised and corrected accordingly. For example, the correct one is "kg" not "Kg".

Author Response

Dear Referee, we would like to thank the referees for their suggestions, which we have incorporated into the text. We believe that their valuable contributions have enriched the work.

Reviewer 3 Report (New Reviewer)

I have carefully read your manuscript entitled: "Status of Ecosystem Services in abandoned mining areas. Management proposal". In my opinion, the paper is fascinating and valuable. Reporting the state of the environment, especially the environment changed by anthropogenic activity, is very important and valuable for potential readers dealing with various fields of science. What is more, I really like the figures in this manuscript.

- line 7 - please add the entire SW full name and abbreviation in brackets; the same in the regular text when you use it the first time; after that, only the abbreviation in the following sentences;

- I suggest "Retama sphaerocarpa" and "phytoremediation" addition to the keywords;

- please add the full names of chemical elements when you write about them the first time and add symbols in brackets; after that, you can use only symbols (except for the first word in a sentence - then you should always write the full name, not the symbol);

- line 32 - There is no supplementary file ((Supplementary Material) in my version of the manuscript submitted for review;

- line 90 - please add information on which parts of the environment were studied;

- Figure 3, line 236 - please add the full name of NGR, sometimes potential reader starts reading the manuscript from the figures and tables;

- Figure 6, line 255 - "kg" instead "Kg";

Best regards,

Reviewer

Author Response

Dear Referee, we would like to thank the referees for their suggestions, which we have incorporated into the text. We believe that their valuable contributions have enriched the work.

Reviewer 4 Report (New Reviewer)

Dear Authors, your manuscript „Status of ecosystem services in abandoned mining areas. Management proposals” is an goog example of very interesting and well written scientific report where valuable data were properly interpreted and right conclusions were drawn. Results are presented in a clear way and in my opinion Figure 4 has the highest value. I am certain that this manuscript is within the scope of the journal and Special Issue. I have just few remarks and recommendations. Please describe analytical methods in a more detailed way. For example meta(loid)s were determined in plant tissues (Figure 6) but no method is mentioned. Also description of methods used in analysis of soil is in my opinion too short. There are some technical errors for example in head of  Table 1 instead of g/kg is gr/kg and instead od cmol/kg is Cmol/kg or in Figure 6 on all axis there is mg/Kg instead of mg/kg. Small problem is in line 299 where Author name was given without number of reference. Also in Reference section in references from number 58 to 72 year of publication is not given in bold. Despite of this really small drawbacks I would like to confirm that I evaluate your manuscript as a valuable input both from scientific and practical aspects.

Author Response

Dear Referee, we would like to thank the referees for their suggestions, which we have incorporated into the text. We believe that their valuable contributions have enriched the work.

Round 2

Reviewer 1 Report (New Reviewer)

Dear authors I appreciate the hard work done

in this last version. I think the manuscript is able to be published without other revisions.

Congratulation,

Reviewer

Reviewer 2 Report (New Reviewer)

I think the authors have made a significant effort in response to the review comments and have addressed the majority of issues raised.

Still, I keep thinking that title is not informative about the goals or results of the study. The title promises more than what the manuscript delivers. In this sense, in addition to not being rigorous, it could be a disappointment for readers....

This manuscript is a resubmission of an earlier submission. The following is a list of the peer review reports and author responses from that submission.

Round 1

Reviewer 1 Report

The manuscript entitled «Phytoremediation as a Biotechnological Alternative for the Remediation of Ecosystem Services in Abandoned Mining Areas» is scientifically interesting and original.

The manuscript is well written and in scientific language and style.

However, there are some points in the manuscript that need improvement:

1.     In my opinion the words in the title of the paper should not be repeated in the keywords. So the words “ecosystem services” should be deleted or replaced by other words or phrases.

2.     As and Sb are not metals!!!! They are metalloids which means that they are non-metallic elements that have metallic chemical behavior. The authors should change all the points (phrases or words) in the text and in the title, as they have studied metals and metalloids.

3.     I think it would be useful for the explanation of the results to refer to the soil taxonomy orders of the soil samples studied. Soil classes or soil orders are of decisive importance for many of the properties related to the retention or not of metal ions in soils. A paragraph needs to be added to the introduction chapter to provide the necessary theoretical background.

Regarding the properties and characteristics of soil taxonomy order, information can be used from “Alloway” s and “Kabata-Pendias” Soil Science books. In addition, the following article (among others) might be useful:

 “Pollution assessment of potentially toxic elements in soils of different taxonomy orders in central Greece. Environ Monit Assess (2019) 191: 106, https://doi.org/10.1007/s10661-019-7201-1”

In my opinion, there is a lack of Soil Science knowledge… CEC and Organic matter are the key factors that determine the behavior of metal cations, in soil solution or the distribution between soil solid phase and soil solution, but soil taxonomy (soil orders) should be discussed further.          

4.      As far as contamination caused by the leaching of metals, in my opinion, there is a small deficit regarding the determination of soil moisture, but also the possibility of calculating the percentage of water that can be washed into the deeper layers of the soil profile. The information provided in the article: “Assessment of Contamination Management Caused by Copper and Zinc Cations Leaching and Their Impact on the Hydraulic Properties of a Sandy and a Loamy Clay Soil”, Land, 2022, 11(2), 290”, (among others) might be useful.

5.     An extensive report on the detection limits of the measured elements needs to be made in the “Materials and Methods” chapter. Also, if a certified sample was used, recovery rates for each element should be reported.

Author Response

In my opinion the words in the title of the paper should not be repeated in the keywords. So the words “ecosystem services” should be deleted or replaced by other words or phrases. 

A. Corrected

2. As and Sb are not metals!!!! They are metalloids which means that they are non-metallic elements that have metallic chemical behavior. The authors should change all the points (phrases or words) in the text and in the title, as they have studied metals and metalloids..

A. Corrected

3.I think it would be useful for the explanation of the results to refer to the soil taxonomy orders of the soil samples studied. Soil classes or soil orders are of decisive importance for many of the properties related to the retention or not of metal ions in soils. A paragraph needs to be added to the introduction chapter to provide the necessary theoretical background.

Regarding the properties and characteristics of soil taxonomy order, information can be used from “Alloway” s and “Kabata-Pendias” Soil Science books. In addition, the following article (among others) might be useful:

 “Pollution assessment of potentially toxic elements in soils of different taxonomy orders in central Greece. Environ Monit Assess (2019) 191: 106, https://doi.org/10.1007/s10661-019-7201-1”

In my opinion, there is a lack of Soil Science knowledge… CEC and Organic matter are the key factors that determine the behavior of metal cations, in soil solution or the distribution between soil solid phase and soil solution, but soil taxonomy (soil orders) should be discussed further.  

A.- The study area is characterized by the presence of eutrophic Regosol developed from slates (lines 147-149) and is discussed in lines 294-313. The appreciations referred to this question have been incorporated in the text.

4. As far as contamination caused by the leaching of metals, in my opinion, there is a small deficit regarding the determination of soil moisture, but also the possibility of calculating the percentage of water that can be washed into the deeper layers of the soil profile. The information provided in the article: “Assessment of Contamination Management Caused by Copper and Zinc Cations Leaching and Their Impact on the Hydraulic Properties of a Sandy and a Loamy Clay Soil”, Land, 2022, 11(2), 290”, (among others) might be useful.

A.- The appreciations referred to this question have been incorporated in the text.

5. An extensive report on the detection limits of the measured elements needs to be made in the “Materials and Methods” chapter. Also, if a certified sample was used, recovery rates for each element should be reported.

A.- The detection limits provided by the laboratory have been incorporated into the text.

Reviewer 2 Report

This manuscript studied the heavy metal pollution status around abandoned mining areas. It is interesting and with the scope of this journal. However, there are so many problems in this manuscript. I do not think this manuscript can be accepted at present version. The detailed comments are listed as follows:

1. The author emphasized the ecosystem services several times in the manuscript, but how to quantitatively description about this index? No content about this has been appeared in the manuscript. What is the relationship between heavy metal content and ecosystem service? There are so many elements in heavy metals, does each element have the same impact on ecosystem service?

2. The heavy metal content in the plants is not measured at the present study. How can the author say that the plant can be used for phytoremediation?

3. The introduction is tediously, confused and unclear, it was not written for the content of this study, which results in the readability of this manuscript is very poor.

4. Lines 76 to 96, all these content introduced the function of soil. But it can only be used as the content of science popularization, without any support for the study content.

5. Line 180 to 182, authors analyzed the content of heavy metals in these samples? But why the content of these heavy metals in various plants is not presented in the manuscript?

In general, the title and content of this manuscript do not matched, and the readability is poor, and is unable to be published in the journal.

Author Response

Dear reviewer, we would like to thank you for the dedication and attention you have given to our work. As for the suggestions you have made, all of them have been considered and have been reflected in the text and supplementary material we have provided. They will undoubtedly enrich the study carried out. We would also like to take this opportunity to make some comments that are not included in the text but which we believe will help to understand the decisions taken. And of course, we are open to any suggestions and clarifications that may be required.

  1. The author emphasized the ecosystem services several times in the manuscript, but how to quantitatively description about this index? No content about this has been appeared in the manuscript. What is the relationship between heavy metal content and ecosystem service? There are so many elements in heavy metals, does each element have the same impact on ecosystem service?

A.- Ecosystem Services or Ecosystem Services refer to the resources provided by the ecosystem. According to FAO (https://www.fao.org/ecosystem-services-biodiversity/en/), they are the multitude of benefits that nature provides to society, therefore the quantification of them is very complicated, in fact many are the studies that try to establish an economic value to these benefits or to their absence or deterioration (Boerema et al., 2017). In this sense, concepts such as soil security try to focus on the fact that the deterioration of soil resources, therefore, for example, of pollution, has a negative influence on the provision, regulation, support or culture, these being the main ecosystem services (Durand et al., 2021). We have included a reference to this in the text to make it easier to read and understand. We have included as Supplementary Material references on how different heavy metals affect human health and ecosystems.

  1. The heavy metal content in the plants is not measured at the present study. How can the author say that the plant can be used for phytoremediation?

A.- Data corresponding to the content of metal(loid)s in plants have been incorporated and analyzed in the text.

  1. The introduction is tediously, confused and unclear, it was not written for the content of this study, which results in the readability of this manuscript is very poor.

A.- We have rewritten the parts that in our opinion could be more difficult to express due to the complexity of the concepts to avoid a longer introduction.

  1. Lines 76 to 96, all these content introduced the function of soil. But it can only be used as the content of science popularization, without any support for the study content.

A.- The less scientific part has been rewritten.

  1. Line 180 to 182, authors analyzed the content of heavy metals in these samples? But why the content of these heavy metals in various plants is not presented in the manuscript?

A.- Data corresponding to the metal(loid)s content in plants have been incorporated and analyzed in the text.

Round 2

Reviewer 1 Report

Dear editors In my opinion the authors faithfully followed the suggestions and advice of the reviewers to improve the quality of their manuscript. The manuscript now meets the scientific requirements for publication in the journal.

Reviewer 2 Report

Authors revised this manuscript, but some of the problems become worse. It still cannot be accepted. Some of the problems were listed as follows:

1. The language is a serious problem. There are many mistakes in grammar and most them is unacceptable.

2. Author should delete the description about ecosystem services from the manuscript, the ecosystem service can be calculated by some model.

3. Author added the bioaccumulation factor in the revised manuscript. However, author just uses the metal contents in plant leaves. It is unreasonable. The biomass of the plant leaves is the smallest compared with other tissues.